# The Role of Nitric Oxide in the Growth and Development of *Schizophyllum commune* Under Anaerobic Conditions

**DOI:** 10.3390/microorganisms13040887

**Published:** 2025-04-12

**Authors:** Dongxu Li, Chen Chu, Mengshi Zhao, Suying Hou, Changhong Liu

**Affiliations:** 1State Key Laboratory of Pharmaceutical Biotechnology, School of Life Sciences, Nanjing University, Nanjing 210023, China; laolang_2012@163.com (D.L.); mg1930112@smail.nju.edu.cn (C.C.); 602022300055@smail.nju.edu.cn (M.Z.); 2College of Life Sciences, Yunnan University, Kunming 650500, China; housuying1008@163.com

**Keywords:** nitric oxide, mycelium growth, basidiospore germination, fruiting body formation, anaerobic, *Schizophyllum commune*

## Abstract

Nitric oxide (NO) is a widely recognized signaling molecule found across various organisms, yet its specific effects on fungal growth and development under anaerobic conditions remain underexplored. This study investigates how NO influences the growth and development of *Schizophyllum commune* 20R-7-F01 under anaerobic environments. The results demonstrated an increase in endogenous NO levels during mycelial growth and basidiospore germination. The addition of cPTIO, a NO scavenger, inhibited mycelial growth, delayed basidiospore germination, and reduced the expression of genes involved in basidiospore germination, highlighting the critical role of NO in fungal growth and development. On the other hand, exogenous NO supplementation accelerated mycelial growth and facilitated the formation of primordia, suggesting NO’s potential as a key regulator of fungal development. These findings deepen our understanding of NO’s contribution to fungal growth in anaerobic conditions and offer new perspectives on its role as a signaling molecule in the development of *S. commune* communities, shedding light on the metabolic regulation of anaerobic microorganisms.

## 1. Introduction

*Schizophyllum commune*, a basidiomycetous fungus, is widely recognized for its remarkable ecological adaptability, particularly its ability to thrive in diverse environmental conditions [1,2]. As a white-rot fungus [3], *S. commune* plays a crucial role in nutrient cycling and decomposition within natural ecosystems [4]. Beyond its ecological importance, it has also gained significant attention in biotechnological research due to its unique enzymatic capabilities, especially in the degradation of lignocellulosic materials [5,6]. These features make it an ideal model organism for studying fungal physiology and growth patterns.

Nitric oxide (NO), a small and diffusible molecule [7], plays essential roles in numerous biological processes in multicellular organisms, such as cell signaling, immune response, and vasodilation [8,9,10]. Recent studies, however, have demonstrated that NO functions diversely in fungi as well. In species like *Fusarium sulphureum*, NO strong inhibits mycelial growth [11], and, in *Pleurotus ostreatus*, it negatively regulates primordia formation by inhibiting mitochondrial aconitase [12]. In *Saccharomyces cerevisiae*, NO has been shown to protect cells from heavy metal toxicity, heat shock, and high hydrostatic pressure, while also influencing metabolic processes through the transcription factor Ace1 [13,14].

NO has also been implicated in various stages of fungal development, such as conidia formation in *Neurospora crassa* and *Coniothyrium minitans* [15,16,17], conidia germination in *Colletotrichum coccodes* [18], sporangium stalk development in *Phycomyces blakesleeanus* [19], the development of ascospores in *Schizosaccharomyces pombe* [20], and zoospore development in *Blastocladiella emersonii* [21]. In the wheat stripe rust pathogen, *Puccinia striiformis* f.sp. *tritici*, NO regulated the polar growth of germ tubes [22]. However, these studies were all conducted under aerobic conditions, and the role of NO in fungal growth and development under anaerobic conditions remains unclear.

The oxygen concentration affects the morphology of *S. commune*, with high concentrations of oxygen (0.5%) forcing fungal cells to change their morphology by reducing the diameter of the hyphae and forming spicules on the surface of the hyphae [23]. Under anaerobic conditions, fungi like *S. commune* must adjust their metabolic strategies to survive in the absence of oxygen, relying on anaerobic respiration, fermentation, or alternative metabolic pathways [24,25]. However, it is not yet understood whether this metabolic shift is influenced by NO or whether NO can act as an effective signaling molecule in such environments. Therefore, exploring the role of NO in the growth and development of *S. commune* under anaerobic conditions is crucial to understanding how this fungus adapts to and thrives in environments with a limited oxygen supply.

The primary aim of this study was to explore the role of NO in the growth and development of *S. commune* 20R-7-F01 under anaerobic conditions. As a dominant fungus in anaerobic subseafloor sediments, *S. commune* possesses unique anaerobic adaptation mechanisms that allow it to thrive in anaerobic environments. Our findings suggest that under anaerobic conditions, NO not only supports the vegetative growth of the mycelium and the germination of basidiospores but also plays a crucial role in the initiation of sexual reproduction. These results highlight the significance of NO as a key signaling molecule for fungal growth and development in anaerobic environments, providing new insights into the adaptive strategies of fungi in anaerobic habitats.

## 2. Materials and Methods

### 2.1. Fungal Strain and Culture Media

The strain *Schizophyllum commune* 20R-7-F01 (CGMCC 5.2202) used in this study was isolated from sediments 1966 m below the seafloor during IODP Expedition 337 [26]. Detailed information about the habitat and culture conditions for this strain has been previously described [2,23]. The minimal medium (MM) contained glucose (C_6_H_12_O_6_, 20 g L^−1^), asparagine (C_4_H_8_N_2_O_3_, 1.5 g L^−1^), dipotassium hydrogen phosphate (K_2_HPO_4_, 1.0 g L^−1^), potassium dihydrogen phosphate (KH_2_PO_4_, 0.46 g L^−1^), magnesium sulfate heptahydrate (MgSO_4_·7H_2_O, 0.5 g L^−1^), vitamin B1 (0.12 mg L^−1^), and ferric chloride hexahydrate (FeC1_3_·6H_2_O, 5 mg L^−1^). Trace elements were prepared according to the Whitaker method [27]. All the chemicals were purchased from Sigma-Aldrich (St. Louis, MO, USA).

### 2.2. Mycelial, Basidiospore, and Fruiting Body Cultures of S. commune 20R-7-F01 Under Anaerobic Conditions

Mycelial culture: Mycelia were cultured in liquid MM for 72 h, followed by washing 2–3 times with sterile deionized water. Subsequently, 0.5 g (wet weight) of the mycelial biomass was transferred to anaerobic bottles (Exetainer; Labco, London, UK) containing 15 mL of liquid MM, which was sealed (without headspace). The bottles were supplemented with either 10 µM sodium nitroprusside (SNP, a NO donor) [28] or 200 µM 2-(4-carboxyphenyl)-4,4,5,5-tetramethylimidazoline-oxyl-3-oxide (cPTIO, a NO scavenger) [29,30] and incubated anaerobically at 30 °C for a designated period. The NO levels and biomass accumulation in the mycelium were subsequently measured.

Basidiospore germination: The collected basidiospores were suspended in sterile deionized water at a concentration of 1 × 10^8^ mL^−1^. Then, 1 mL of the basidiospore suspension was transferred to an anaerobic bottle containing 15 mL of liquid MM (the concentration of basidiospores was now 10^6^ mL^−1^). The suspension was incubated anaerobically at 30 °C for a specified period. The NO levels in the basidiospores, as well as the basidiospore length and the expression of genes associated with basidiospore germination, were assessed.

Fruiting body culture: To evaluate fruiting body development and gene expression related to fruiting body formation, 2 mm diameter mycelial plugs were cultured on solid MM, with or without 10 µM SNP, at 30 °C for 168 h. Following this, the culture was exposed to 240 h of light treatment (12 h day/12 h night).

All the experiments were performed under anaerobic conditions, and mycelial culture and basidiospore germination were performed in liquid culture medium. The medium in each anaerobic bottle was purged with nitrogen. The specific time of nitrogen purging was determined by the change of the liquid MM (supplemented with 1 µM Resazurin (Thermo Fisher Scientific, Waltham, MA, USA) as an anaerobic indicator [31]) from blue-purple to colorless. The fruiting bodies were cultured in a culture dish containing a solid culture medium. The culture dish was placed in an anaerobic culture tank and purged with nitrogen for 1 h to ensure an anaerobic environment, and Oxoid™ Resazurin (Thermo Fisher Scientific) was attached to the inner wall of the culture tank to maintain an anaerobic culture environment [32].

### 2.3. Basidiospore Collection from Fruiting Bodies

To collect basidiospores from *S. commune*, mature fruiting bodies were first induced under aerobic conditions [23]. A 2 mm diameter mycelial plug was inoculated on solid MM and cultured at 30 °C for 120 h. Following this, the culture was subjected to a light treatment of 120 h (12 h day/12 h night) to stimulate fruiting body formation. (After culturing for different time periods, we observed that, at this time point, the growth of the mycelium was at a stable stage and had the potential to induce fruiting body formation. The time setting of light treatment is to ensure that the fungus can fully respond to environmental changes and form fruiting bodies under the influence of light induction. Primordia began to appear at 24 h of light treatment; the fruiting bodies appeared at 48 h and matured at 120 h). After fruiting bodies developed, a new sterile culture dish was prepared by fixing two sterile syringe needles on the inner wall of the culture dish cover using glue. When the fruiting body structure is intact, the cap is unfolded, and the gills are clearly visible, we consider the fruiting body to be mature. At this time, the morphology of the basidiospores was observed under a microscope, and we started collecting basidiospores after confirming the integrity. The mature stipe caps were carefully separated, and the stipe was fixed onto the syringe needles, ensuring that the gills faced downward to allow the basidiospores to fall off and settle naturally. The culture dishes were then covered and left standing for 4–24 h. After the incubation period, the culture dish cover was removed, and the lower surface of the cover was gently rinsed with sterile water to collect the basidiospores.

### 2.4. Qualitative Detection of NO Concentration in Mycelia and Basidiospores

The NO concentration in *S. commune* mycelium was measured using the specific NO fluorescent probe DAF-2DA (4,5-diaminofluorescein diacetate, Sigma-Aldrich) as described by Wang et al. [33]. DAF-2DA is a cell-permeable, lipophilic compound that is non-fluorescent until hydrolyzed by cytosolic esterases into the weakly fluorescent DAF-2. In the presence of NO radicals, DAF-2 is further converted into the highly fluorescent triazole derivative, DAF-2T. The fluorescence intensity of DAF-2T correlates with the intracellular NO concentration.

For the measurement, the mycelia and basidiospores were incubated with 2.5 μM DAF-2DA at 37 °C for 20 min. After incubation, the samples were washed 3–5 times with sterile deionized water and placed on microscope slides. The NO concentration at the leading edge of the mycelium was observed using an Olympus BX53 fluorescence microscope (Evident, Tokyo, Japan) with excitation at 450–490 nm and emission at 500–530 nm. The fluorescence intensity was quantified using ImageJ (1.51n) software.

### 2.5. Quantification of NO Content

For the quantitative analysis of NO content, the mycelia were washed 2–3 times with deionized water and then homogenized with 1 mL of a 50 mM acetic acid solution (pH 3.6) containing 4% (*w*/*v*) zinc acetate. The homogenate was centrifuged at 9500× *g* for 15 min at 4 °C. The NO content was measured following the manufacturer’s protocol (A013-2-1, Nanjing Jiancheng Institute of Biological Engineering, Nanjing, China). The protein concentration was determined using the Bradford method [34]. Absorbance readings were recorded using an Infinite M200 Pro microplate reader (Tecan, Research Triangle Park, Durham, NC, USA).

### 2.6. Gene Expression Analysis Using RT-qPCR

Gene expression levels were assessed using real-time polymerase chain reaction (RT-qPCR) [35]. Total RNA was extracted from 100 mg of mycelia or basidiospores, which were harvested by centrifugation, using RNAiso Plus (TaKaRa, Da Lian, China) according to the manufacturer’s protocol. Reverse transcription was performed using the PrimeScript RT Reagent Kit, which contains a gDNA Eraser (Takara), in strict accordance with the provided instructions. Quantitative PCR analysis of the target genes was carried out using SYBR Premix Ex Taq II (TaKaRa). Relative gene expression was calculated using the 2^−ΔΔCT^ method [36], with *Actin* as the reference gene. The primers used for gene expression quantification are detailed in Appendix A.

### 2.7. Statistical Analysis

All the experiments were conducted with a minimum of three biological replicates. Data are presented as the mean ± standard error. Statistical analysis was performed using SPSS version 28.0.1.1 (Statistical Package for Social Sciences) and ImageJ software. Analysis of variance (ANOVA) was used to compare differences between multiple treatment groups. Subsequently, Tukey’s test (for homogeneity of variance between groups) and the least significant difference (LSD) test (for unequal variance or when a more sensitive test is needed) were used for pairwise comparisons to assess significant differences between different treatments. The significance level for all statistical analyses was set at *p* ≤ 0.05 [37].

## 3. Results

### 3.1. Dynamics of NO Levels During the Growth of S. commune Mycelium

We utilized the NO-specific fluorescent probe DAF-2DA (capable of producing strong fluorescent signals through high selectivity) to investigate the changes in NO content during the growth of *S. commune* (Figure 1A,B). The results showed that, compared with that at 0 h, the NO fluorescence intensity gradually increased from 4 to 16 h, with the NO content rising from 2.60 to 5.74 μmol mg^−1^ protein. After 16 h, the NO intensity began to decline and stabilized by 24 h, showing a 23% increase in fluorescence intensity and a 30% rise in NO content compared to the 0 h time point (Figure 1). Throughout the culture period, the trend in NO content mirrored that of NO fluorescence intensity (Figure 1B,C), with a strong positive correlation observed between the two (Appendix A). These findings suggest that the NO levels in the mycelium increase throughout the growth of *S. commune*. Fluctuations in NO may act as a molecular switch during fungal growth, regulating the activities of key enzymes related to the cell cycle, redox state, and metabolic pathways.

At the same time, we also analyzed the temporal trend of nitric oxide (NO) content in *S. commune* fungi under aerobic conditions. Under aerobic conditions, the trend of NO content was the same as that under anaerobic conditions. It gradually increased after the start of cultivation (3.44 μmol mg^−1^ protein), reached a peak value (8.27 μmol mg^−1^ protein) at about 12 h, and then gradually decreased and remained stable (3.76 μmol mg^−1^ protein); however, the NO content was always higher than that under anaerobic conditions (Appendix A). This shows that *S. commune* can quickly accumulate NO in an aerobic environment, which may be related to its physiological process of adapting to an oxygen-rich environment. Meanwhile, under anaerobic conditions, the NO accumulation is lower, reflecting its adaptation mechanism to a low-oxygen environment.

### 3.2. NO Dynamics During Basidiospore Germination in S. commune

The germination process of basidiospores involves several stages: activation, isotropic growth, and polarized growth. Initially, basidiospores undergo activation, triggering the germination process. This is followed by isotropic growth, which is marked by swelling and water uptake, along with wall growth. Polarized growth begins once wall deposition becomes polarized, forming a germ tube [38].

When *S. commune* basidiospores were inoculated in 15 mL MM liquid and incubated, the NO levels gradually increased during the first 10 h of incubation. The NO levels peaked at 10 h and then stabilized between 12 and 18 h (Figure 2A,B). Our observation indicated that the activation phase occurred within the first 10 h, with basidiospores entering the isotropic growth phase after 10 h. Polarized growth began at 18 h (Figure 2C). These findings demonstrate that NO levels rise throughout the germination process of *S. commune* basidiospores.

### 3.3. Effect of the NO Scavenger cPTIO on Mycelial Growth and Basidiospore Germination in S. commune

To investigate the role of NO in mycelium growth and basidiospore germination, we treated *S. commune* with the NO scavenger cPTIO (2-(4-carboxyphenyl)-4,5-dihydro-4,4,5,5-tetramethy-1H-imidazolyl-1-oxy-3-oxide). As shown in Figure 3A, after 96 h of mycelial growth in liquid MM, the treatment with cPTIO significantly reduced the net accumulation of mycelial biomass by 29% compared with that in the control group without cPTIO. We speculate that the inhibitory effect of cPTIO may affect the development of *S. commune* by interfering with NO-mediated signaling pathways.

Additionally, when the basidiospores were cultured in 15 mL MM liquid, the exogenous addition of cPTIO delayed the peak NO levels in the basidiospores by 6 h (Figure 3B). Furthermore, the time of the isotropic and polarized growth phases of the basidiospores, which typically occurs between 12 and 18 h, was also extended by 6 h (from 18 to 24 h) when treated with cPTIO (Figure 3C). This is because NO promotes the early growth and division of basidiospores by activating specific signaling pathways (e.g., cAMP pathway, MAPK pathway) during fungal development. When NO is removed by cPTIO, the NO level in the basidiospores decreases, and the progression of the cell cycle is delayed, resulting in a prolonged growth phase. These findings suggest that NO scavenging with cPTIO inhibits mycelial growth and delays basidiospore germination.

### 3.4. Endogenous NO Regulation of Basidiospore Germination Genes in S. commune

To explore the role of endogenous NO in the regulation of genes involved in basidiospore germination, we cultured *S. commune* basidiospores in 15 mL of liquid MM at 30 °C until the end of the isotropic growth period and assessed the expression levels of key germination-related genes. Basidiospore germination is governed by multiple signaling pathways, and we focused on the expression of genes associated with the 3′-5′-cyclic adenosine monophosphate (cAMP) pathway (related to cell metabolism and proliferation) (Figure 4A), mitogen-activated protein kinase (MAPK) pathway (participates in cell proliferation, differentiation, and response to external stimuli) (Figure 4B), C_2_H_2_-type zinc finger protein (important transcription factor in cells) (Figure 4C), and chitin synthase (remodeling of cell structure) (CHS) genes (Figure 4D).

The RT-qPCR analysis revealed that the addition of 200 μM cPTIO significantly downregulated the expression of 24 genes related to basidiospore germination. This included genes from the cAMP pathway such as *PKAR*, *PAC2-1*, *PAC2-2*, *SCAMP2,* and *PKA1* (Figure 4A); genes from the MAPK pathway including *MAPK*, *MTS50*, *MKP2*, *MPS1*, *HOG1*, *MCK1,* and *MST11* (Figure 4B); four C_2_H_2_-type zinc finger protein genes (*CON7*, *DVRA*, *MTFA-1,* and *MTFA-2*) (Figure 4C); and eight chitin synthase genes (*CHS1*, *CHS2*, *CHS3-1*, *CHS3-2*, *CHS4*, *CHS6*, *CHS7*, and *CHS8*) (Figure 4D). By regulating the expression of these genes, NO helps regulate the structure and physiological state of the basidiospores and promotes their transition from dormancy to active germination. These results highlight that NO acts as a regulatory factor in these processes, participating in the regulation of signal transduction and gene expression, and illustrate the important role of NO in regulating basidiospore germination.

### 3.5. Influence of Exogenous NO on Mycelial Growth and Sexual Reproduction in S. commune

To examine the impact of exogenous NO on fungal mycelial growth, *S. commune* was cultured in 15 mL of liquid MM containing 10 μM sodium nitroprusside (SNP), a NO donor. The control group was treated with MM medium without SNP. Over the course of the culture, mycelial biomass accumulation increased in both groups, with SNP supplementation resulting in a significantly higher accumulation of biomass than that for the control (Figure 5A). This suggests that NO helps fungi maintain their ability to grow and develop in anaerobic environments by adjusting anaerobic physiological responses.

When cultured on solid MM plates under aerobic conditions, the mycelium formed mature fruiting bodies (Appendix A). However, under anaerobic conditions, the mycelium was unable to initiate sexual reproduction (failed to form primordia) (Figure 5B), in accordance with prior studies [23]. Notably, while SNP treatment under anaerobic conditions did not induce the formation of mature fruiting bodies, it did promote the development of primordia (Figure 5B).

The levels of NO were assessed in the mycelium, fruiting bodies, and primordia under the different culture conditions. After the initiation of sexual reproduction, the NO content in the mycelium was significantly lower than in the fruiting bodies and primordia. SNP treatment under anaerobic conditions increased the NO levels in both the mycelium and the primordia, although the levels were still lower than the NO content in mycelium grown under aerobic conditions (Appendix A). This suggests that under anaerobic conditions, NO may promote fungal growth by activating genes related to anaerobic metabolism, such as glycolysis and fermentation pathways. Under aerobic conditions, NO may help fungi better adapt to oxygen-sufficient environments by regulating the redox status, inhibiting oxidative stress responses, and regulating the cell cycle. This change may also indicate the adaptive role of NO in fungi in responding to different environmental stresses.

Moreover, RT-qPCR analysis revealed that SNP treatment significantly upregulated the expression of several genes related to fruiting body formation, including *Hom1*, *Hom2-1*, *Hom2-2*, *Fst3*, *Fst4*, *Wc-1*, *Wc-2*, *Gat1*, *Bri1*, and *C2h2*, when compared with the control group (Figure 5C). This indicates that NO may promote the expression of these genes by directly or indirectly acting on the promoter regions of these genes or by regulating the activity of transcription factors. The mechanism behind these transcriptional changes may be closely related to the NO-regulated signaling network and the activation of transcriptional regulatory factors. These transcriptional changes suggest that exogenous NO enhances the expression of key genes involved in fruiting body development, thereby facilitating the initiation of primordium formation.

## 4. Discussion

In this study, we demonstrate that NO plays an essential role in mycelium extension, biomass accumulation (Figure 1), basidiospore germination (Figure 2 and Figure 3), and fruiting body formation (Figure 5) in *S. commune* 20R-7-F01 under anaerobic conditions. A key finding from this research was the dynamic pattern of NO accumulation during different stages of fungal growth. The NO levels gradually increased during the early stages of mycelium growth, peaked around 16 h, and then stabilized as growth progressed (Figure 1). This temporal variation in NO concentrations suggests that NO is involved in promoting early growth processes, such as mycelium extension and nutrient uptake, under anaerobic conditions. This observation aligns with findings from studies on *N. crassa* grown under aerobic conditions, where NO is similarly important for early mycelium development [28]. The subsequent stabilization of NO levels might indicate that the fungus enters a phase focused more on maintaining growth rather than on rapid expansion. The effects of NO on mycelium growth were further confirmed by treatments with SNP (a NO donor) (Figure 3) and cPTIO (a NO scavenger) (Figure 5), which showed that NO positively influenced mycelium growth. While this has been established for *N. crassa* under aerobic conditions [39], our study is the first to confirm this effect of NO on fungal mycelium growth under anaerobic conditions.

The germination of fungal spores is generally considered an aerobic process, with previous studies not investigating whether NO levels fluctuate during this process [40,41]. Our study provides the first evidence that the basidiospores of *S. commune* can germinate under anaerobic conditions (Figure 2), and we observed a time-dependent change in NO levels during germination. The NO levels peaked at 10 h and then stabilized (Figure 2). As the NO concentrations changed, we observed changes in the spore morphology, such as a transition from a quiescent state to an actively growing state, as evidenced by the initial expansion of the hyphae. Therefore, changes in NO levels are a key indicator of changes in spore physiological state and morphology. When cPTIO was introduced, the activation time of the basidiospores was delayed from 10 to 16 h, although the isotropic growth period and the initiation of polarized growth were unaffected (Figure 3). While cPTIO did not completely inhibit basidiospore germination, it is possible that the concentration of cPTIO used was lower than that in previous studies [28], which may explain the partial effect observed. Additionally, gene expression analysis revealed that the genes associated with spore germination signaling were downregulated upon NO inhibition (Figure 4).

This temporal fluctuation of NO during basidiospore germination suggests that NO plays an essential role in the activation step, a critical process for successful spore germination. In the early stages of fungal development, NO first regulates the cell cycle by activating the cAMP pathway, MAPK pathway, and C_2_H_2_-type zinc finger proteins. By regulating the expression of cell cycle-related genes, it ensures that the basidiospores can enter the germination stage at the appropriate time. In the activation stage, NO also participates in regulating cell wall synthesis (chitin synthase). NO promotes the transition of spores from dormancy to germination by inducing growth- and development-related genes and enhancing the expression of these genes. Our results are consistent with previous studies showing that NO regulates spore germination in other fungi, such as *C. coccodes* and *P. striiformis* f.sp. *tritici*, under aerobic conditions [42].

The role of NO in fungal growth and reproduction is multifaceted, and while it positively influences both mycelial growth and basidiospore germination, its involvement in fruiting body formation appears to be more complex. Under anaerobic conditions, control cultures of *S. commune* failed to initiate fruiting body formation (Figure 5), consistent with previous studies suggesting that many fungi are unable to form fruiting bodies in anaerobic environments [23]. However, treatment with SNP (a NO donor) successfully initiated the fruiting body formation process, resulting in the formation of primordia. Additionally, the expression of genes related to fruiting body development was significantly upregulated at this stage (Figure 5). This contrasts with findings from studies conducted under aerobic conditions, where NO negatively regulated the formation of primordia in *P. ostreatus* by inhibiting mitochondrial gene expression [12].

While we did not observe complete fruiting body formation in our study, the formation of primordia indicates that NO plays a critical role in the initiation of fruiting body formation and early differentiation. These findings suggest that, while NO is essential for triggering the early stages of fruiting body formation, other factors, potentially including a specific threshold of NO, may be necessary to complete the process. Under anaerobic conditions, complete fruiting body formation was still not achieved when SNP was used, which may be related to the concentration of NO used. The effects of different concentrations of SNP on fungal development can be further explored to investigate whether there is a more suitable SNP concentration that allows fruiting bodies to form completely. Of course, the effect of NO signaling under anaerobic conditions may also be regulated or interfered with by other signaling molecules (such as H_2_O_2_ and Ca^2+^), leading to the absence or incomplete formation of fruiting bodies. Future studies can explore whether there is regulation or interaction between NO and these signaling molecules, so as to obtain a more comprehensive understanding of the effect of NO on fruiting body formation.

Overall, the results of this study underscore the pivotal role of NO as a signaling molecule in regulating various stages of fungal development, especially under anaerobic conditions. They also suggest that the role of NO is not specific to anaerobic processes. NO’s ability to modulate gene expression in response to both endogenous and exogenous stimuli highlights its importance in coordinating cellular responses to environmental changes. Moreover, the interplay between NO and other signaling pathways, such as the cAMP and MAPK pathways, suggests that NO serves as a key mediator that integrates various cellular responses to environmental cues. These findings provide valuable insights into how *S. commune* and other fungi adapt to anaerobic environments, contributing to a deeper understanding of fungal biology and the adaptive mechanisms fungi employ to thrive in such conditions.

## Figures and Tables

**Figure 1 microorganisms-13-00887-f001:**
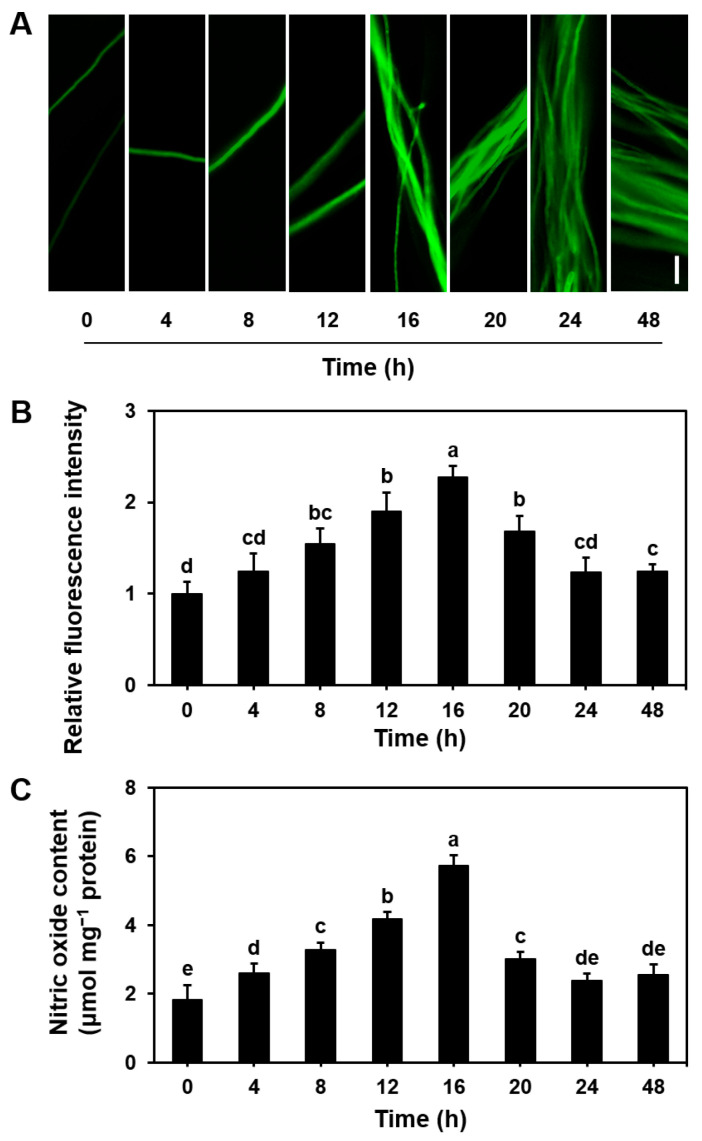
Endogenous nitric oxide (NO) levels during the mycelial growth of *Schizophyllum commune* 20R-7-F01. (**A**) Image showing NO concentration in the mycelium, measured using the NO-specific fluorescent probe DAF-2DA; (**B**) relative fluorescence intensity of NO in the image; (**C**) NO content. The mycelium was cultured in 15 mL of liquid MM at 30 °C for 48 h. The NO content is expressed as μmol mg^−1^ protein. Data are presented as mean ± S.E ((**B**) *n* = 20; (**C**) *n* = 3). Different letters indicate significant differences (*p* ≤ 0.05). Scale bar = 10 μm.

**Figure 2 microorganisms-13-00887-f002:**
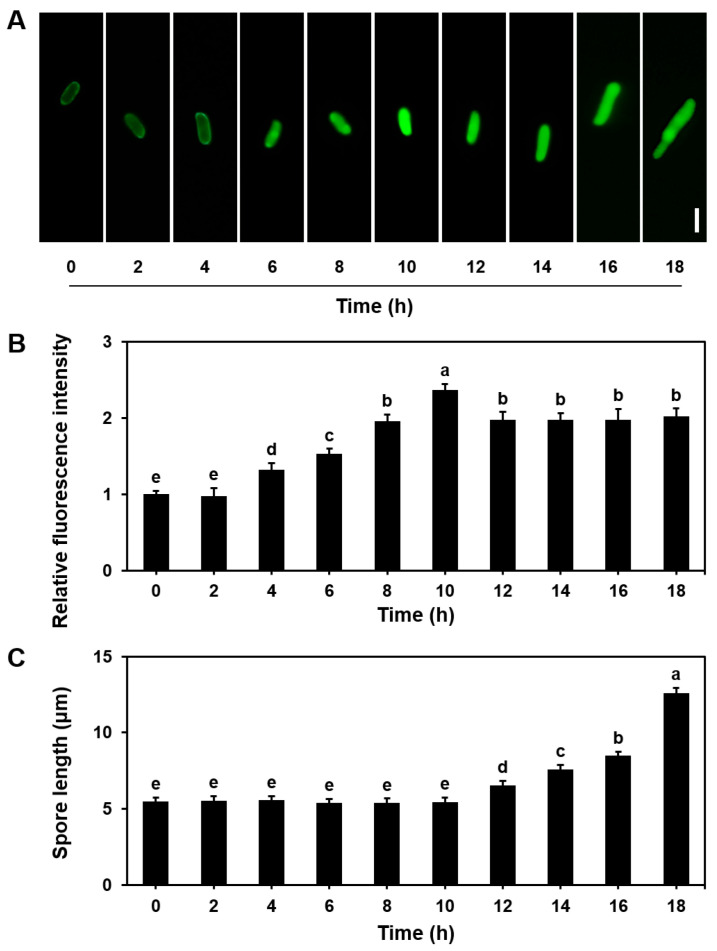
NO levels during basidiospore development of *S. commune*. (**A**) Images showing NO concentration in basidiospores, measured using the NO-specific fluorescent probe DAF-2DA; (**B**) relative fluorescence intensity of NO in the images; (**C**) basidiospore length measured using ImageJ. The basidiospores were cultured in 15 mL of liquid MM at 30 °C for 18 h. Values are presented as mean ± S.E (*n* = 20). Different letters indicate statistically significant differences (*p* ≤ 0.05). Scale bar = 5 μm.

**Figure 3 microorganisms-13-00887-f003:**
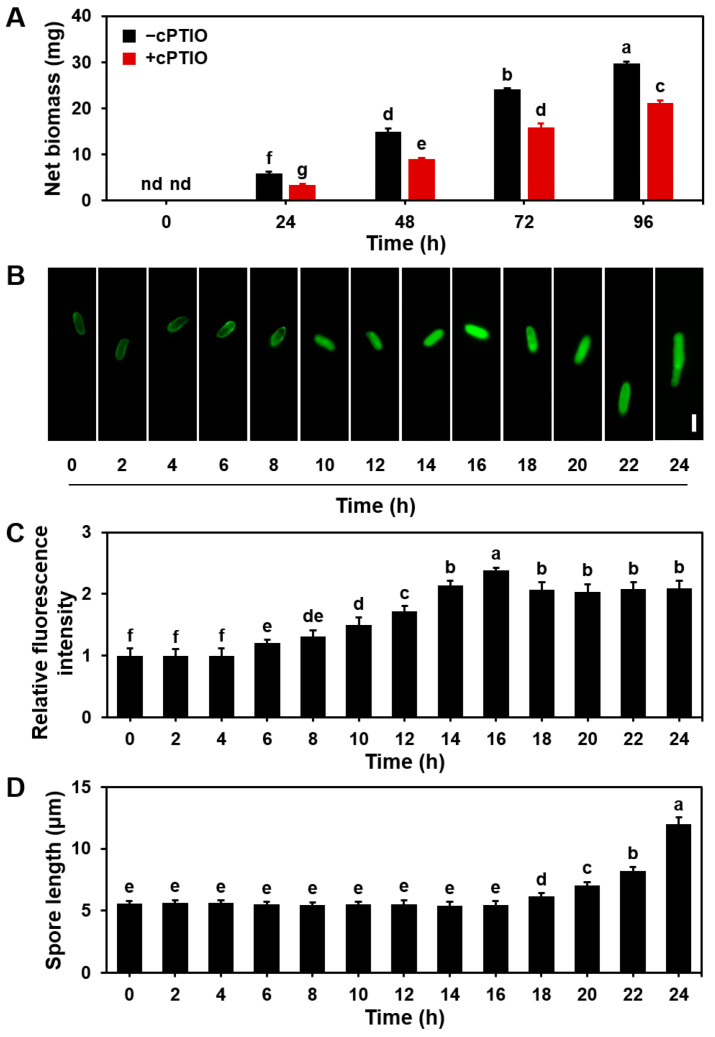
Effect of NO on mycelial growth and basidiospore germination in *S. commune*. (**A**) One gram of mycelium was cultured in 15 mL MM liquid with or without 200 μM cPTIO (NO scavenger) at 30 °C for 96 h. The culture without cPTIO served as the control. (**B**) Images showing NO concentration in basidiospores, measured using the NO-specific fluorescent probe DAF-2DA. (**C**) Relative fluorescence intensity of NO in the images. (**D**) Basidiospore length was measured using ImageJ after culturing the basidiospores in 15 mL of liquid MM at 30 °C for 24 h. Values represent the mean ± S.E ((**A**) *n* = 3; (**B**) *n* = 20; (**C**) *n* = 20; (**D**) *n* = 20). Different letters indicate significant differences (*p* ≤ 0.05). Scale bar = 5 μm. nd = below detection limit.

**Figure 4 microorganisms-13-00887-f004:**
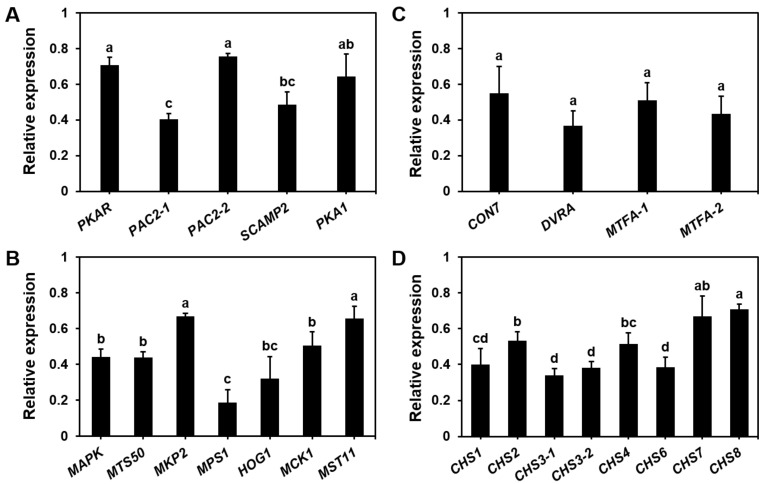
Impact of NO scavengers on the expression of genes associated with basidiospore germination in *S. commune*. (**A**) Genes involved in the 3′-5′-cyclic adenosine monophosphate (cAMP) signaling pathway: *PKAR*, *PAC2-1*, *PAC2-2*, *SCAMP2,* and *PKA1*; (**B**) genes related to the mitogen-activated protein kinase (MAPK) pathway: *MAPK*, *MTS50*, *MKP2*, *MPS1*, *HOG1*, *MCK1,* and *MST11*; (**C**) genes encoding C_2_H_2_-type zinc finger proteins: *CON7*, *DVRA*, *MTFA-1,* and *MTFA-2*; (**D**) genes associated with chitin synthase (CHS): *CHS1*, *CHS2*, *CHS3-1*, *CHS3-2*, *CHS4*, *CHS6*, *CHS7*, and *CHS8*. Basidiospores were cultured in 15 mL liquid MM with or without 200 μM cPTIO at 30 °C for 16 or 22 h. The culture without cPTIO served as a control. Data are expressed as mean ± S.E (*n* = 3). Different letters indicate significant differences (*p* ≤ 0.05).

**Figure 5 microorganisms-13-00887-f005:**
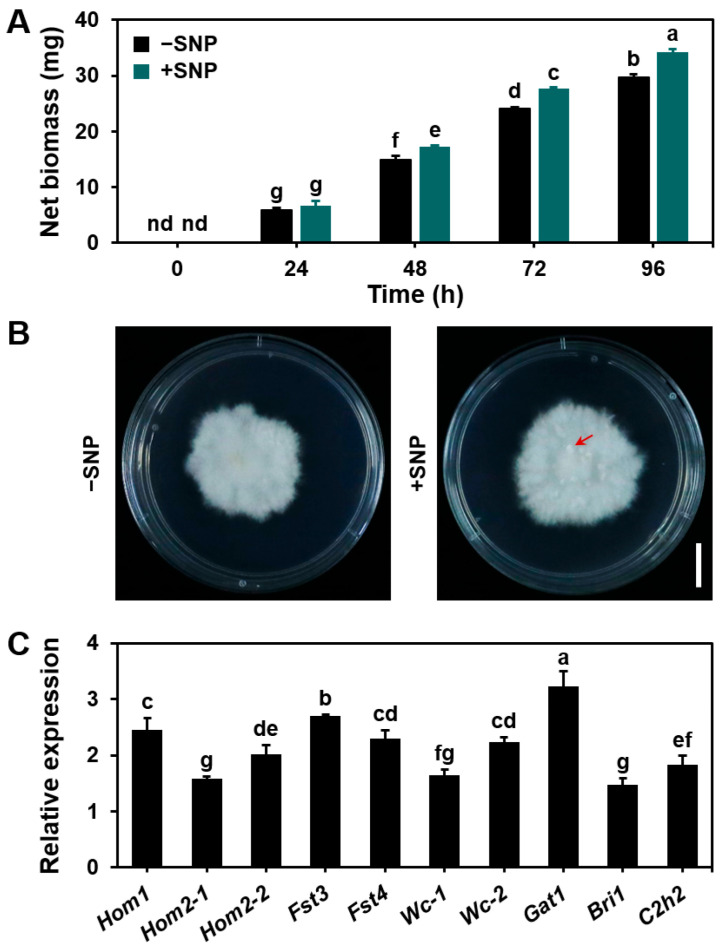
Effects of exogenous NO on mycelial growth, fruiting body formation, and related gene expression in *S. commune*. (**A**) One gram of mycelium was cultured in 15 mL of liquid MM with or without 10 μM SNP (a NO donor) at 30 °C for 96 h. (**B**) Images of culture dishes showing the absence of fruiting bodies and the presence of primordia (arrows). Mycelial plugs (2 mm diameter) were cultured on solid MM with or without 10 µM SNP for 168 h, followed by 240 h of light treatment (12 h day/12 h night). In the +SNP treatment, primordia appeared at 72 h of light treatment, grew between 72 and 120 h, and did not develop further from 120 to 240 h. (**C**) Changes in the expression levels of genes related to fruiting body formation. Gene expression levels in mycelia of two plates in (**B**) were analyzed after exposure to light for 72 h (12 h day/12 h night). The culture without SNP served as the control. Data are presented as mean ± S.E (*n* = 3). The structure pointed by the red arrow is the primordial. Different letters indicate significant differences (*p* ≤ 0.05). nd = below detection limit. Scale bar = 1 cm.

## Data Availability

The original contributions presented in this study are included in the article/Appendix A. Further inquiries can be directed to the corresponding author.

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
