# Peer review of "The Role of Nitric Oxide in the Growth and Development of *Schizophyllum commune* Under Anaerobic Conditions"

_microorganisms, 2025, doi:10.3390/microorganisms13040887_

Round 1

Reviewer 1 Report

Comments and Suggestions for Authors

Using a pharmacological approach, Li et al. observed that a NO donor increases the mycelium development and basidiospore germination rate of Schizophyllum commune; in contrast, supplementation with a NO scavenger decreases the same parameters. They propose that NO's role in S. commune under anaerobiosis is important during vegetative growth and sexual reproduction. PKA and MAPK pathways could regulate those biological processes through NO signaling. The manuscript is well-written and well-structured.

Regardless, I have some comments.

Major comments:

  1. Since oxygen is a morpho regulator for other fungi, please provide details in the material and method section about the conditions used to create anaerobiosis and specify whether all the experiments were performed under those conditions. Also, please add if oxygen concentration affects S. commune morphology in the introduction section.
  2. The germination rate depends on the initial inoculum for fungal spores. A higher initial inoculum spore germination could be inhibited. For other fungi, the concentration of spores in cultures oscillates between 5x105 to 1x107 spores/mL. How did the authors choose the concentration of basidiospores for germination assays?
  3. How did the authors choose the concentration of SNP and cPTIO. Is the accumulation of NO dependent on the concentration of these compounds in S. commune under anaerobiosis?
  4. Please specify the type of analysis used for RT-qPCR assays in the material and methods section. It seems to me like a ∆∆CT analysis. However, how did you normalize the data with the actin and control expressions?
  5. The authors present results where NO is accumulated in aerobic and anaerobic conditions (Figure S2B). Please clarify in the discussion section if the role of NO is specific during anaerobiosis.
  6. In other fungi, metabolic fitness under anaerobic conditions (fermentation) has been described as linked to the carbon source, a fermentable one. If you increase the concentration of this carbon source, you can stimulate anaerobic growth and fermentation. It would also be interesting to analyze whether NO is related to anaerobic metabolism. For example, how is NO accumulation in a medium with four or six % glucose?
  7. Did the authors analyze whether their observations are specific to NO signaling? It would be interesting to include as a control how ROS molecules and a ROS scavenger affected the development of this fungus under anaerobiosis.

Minors’ comments:

The reference figures in lines 159-160 should be Figures 1B and 1C.

Author Response

Comments 1:

Since oxygen is a morpho regulator for other fungi, please provide details in the material and method section about the conditions used to create anaerobiosis and specify whether all the experiments were performed under those conditions. Also, please add if oxygen concentration affects S. commune morphology in the introduction section.

Response 1:

We did not state this clearly, thank you for pointing this out and we agree with this comment. We have changed the title of 2.2 from “Mycelial, basidiospore, and fruiting body cultures of S. commune 20R-7-F01” to “Mycelial, basidiospore, and fruiting body cultures of S. commune 20R-7-F01 under anaerobic conditions” to make it clearer.

We have also supplemented the Materials and methods section with detailed information on the anaerobic conditions used in the experiments and emphasized that all experiments were performed under anaerobic conditions. The description now reads: All experiments were performed under anaerobic conditions, and mycelial culture and basidiospore germination were performed in liquid culture medium. The medium in each anaerobic bottle was purged with nitrogen. The specific time of nitrogen purging was determined by the change of the liquid MM (supplemented with 1 µM Resazurin (Thermo Fisher Scientific, Waltham, MA, USA) as an anaerobic indicator [31]) from blue-purple to colorless. The fruiting bodies were cultured in a culture dish containing solid culture medium. The culture dish was placed in an anaerobic culture tank and purged with nitrogen for 1 h to ensure an anaerobic environment, and Oxoid™ Resazurin (Thermo Fisher Scientific) was attached to the inner wall of the culture tank to maintain an anaerobic culture environment [32].

We added the effect of oxygen concentration on the morphology of S. commune to the Introduction section. The details are as follows: Oxygen concentration affects the morphology of S. commune, with high concentrations of oxygen (0.5%) forcing fungal cells to change their morphology by reducing the diameter of the hyphae and forming spicules on the surface of the hyphae [23].

Comments 2:

The germination rate depends on the initial inoculum for fungal spores. A higher initial inoculum spore germination could be inhibited. For other fungi, the concentration of spores in cultures oscillates between 5x105 to 1x107 spores/mL. How did the authors choose the concentration of basidiospores for germination assays?

Response 2:

We agree with this comment.

When we selected the concentration of basidiospores for the germination assay, we also controlled the concentration within the range of 5x105 to 1x107 spores/mL, but it was not clearly stated here. What we want to describe is: 1 mL of basidiospore suspension with a concentration of 1×108/mL is transferred to an anaerobic bottle containing 15 mL of liquid MM, and the concentration of basidiospores is 106/mL. Therefore, we have supplemented the description of the concentration of basidiospores here: The collected basidiospores were suspended in sterile deionized water at a concentration of 1×108/mL. One milliliter of the basidiospore suspension was transferred to an anaerobic bottle containing 15 mL of liquid MM (the concentration of basidiospores was now 106/mL).

Comments 3:

How did the authors choose the concentration of SNP and cPTIO. Is the accumulation of NO dependent on the concentration of these compounds in S. commune under anaerobiosis?

Response 3:

Thank you for your comments. We selected the concentrations of SNP and cPTIO based on the concentrations reported so far, and the corresponding references have been added to the Materials and Methods. The current description is: The bottles were supplemented with either 10 µM sodium nitroprusside (SNP, an NO donor) [28] or 200 µM 2-(4-carboxyphenyl)-4,4,5,5-tetramethylimidazoline-1-oxyl-3-oxide (cPTIO, an NO scavenger) [29, 30], and incubated anaerobically at 30 °C for a designated period.

Since SNP and cPTIO are specific donors and scavengers of NO, the accumulation of NO in S. commune under anaerobic conditions is dependent on the concentrations of these compounds.

Comments 4:

Please specify the type of analysis used for RT-qPCR assays in the material and methods section. It seems to me like a ∆∆CT analysis. However, how did you normalize the data with the actin and control expressions?

Response 4:

We agree with this comment. We did not describe it clearly here. What we want to show here is the relative expression level changes of genes, and the data were processed using 2-∆∆CT analysis. We have supplemented the Materials and methods section accordingly. The description now reads: Quantitative PCR analysis of the target genes was carried out using SYBR Premix Ex Taq II (TaKaRa). Relative gene expression was calculated using the 2−ΔΔCT method [36], with Actin as the reference gene.

Comments 5:

The authors present results where NO is accumulated in aerobic and anaerobic conditions (Figure S2B). Please clarify in the discussion section if the role of NO is specific during anaerobiosis.

Response 5:

We agree with this comment. We have clarified in the Discussion section whether the role of NO is specific to anaerobic processes, as described in detail: They also suggest that the role of NO is not specific to anaerobic processes.

Comments 6:

In other fungi, metabolic fitness under anaerobic conditions (fermentation) has been described as linked to the carbon source, a fermentable one. If you increase the concentration of this carbon source, you can stimulate anaerobic growth and fermentation. It would also be interesting to analyze whether NO is related to anaerobic metabolism. For example, how is NO accumulation in a medium with four or six % glucose?

Response 6:

We agree with this comment. As you mentioned, the fermentation process is closely related to the carbon source (especially the fermentable carbon source). In our study, although we mainly focused on the role of NO in mycelial growth, basidiospore germination and fruiting body development, we also noticed the potential relationship between NO accumulation and anaerobic metabolism. The glucose concentration we used in this experiment was 2%. We plan to explore the effects of different glucose concentrations (such as 4% and 6%) on NO accumulation in subsequent studies, and analyze the relationship between this process and anaerobic metabolism.

Comments 7:

Did the authors analyze whether their observations are specific to NO signaling? It would be interesting to include as a control how ROS molecules and a ROS scavenger affected the development of this fungus under anaerobiosis.

Response 7:

We agree with this comment. Although this article does not report whether our observations are specific to NO signaling, we found in another part of our work (Interactions between NO and H2O2 in S. commune under anaerobic conditions) that the observations in this article are not specific to NO signaling, and that the regulation of NO and ROS molecules under anaerobic conditions shows a dynamic equilibrium state. Your comments will also provide ideas for our subsequent research.

Comments 8:

The reference figures in lines 159-160 should be Figures 1B and 1C.

Response 8:

We agree with this comment. The reference figure has been modified to “Figures 1B and 1C”.

Reviewer 2 Report

Comments and Suggestions for Authors

The manuscript presents a well-designed and scientifically relevant study that explores the role of nitric oxide (NO) in fungal development under anaerobic conditions. The study is particularly significant given the ecological and biotechnological importance of Schizophyllum commune and the limited understanding of the role of NO in anaerobic environments, as most previous research has focused on aerobic conditions. The manuscript has several strengths, including its comprehensive experimental design, robust methodologies, and clear presentation of results. However, there are areas where the study could be improved to increase its impact and clarity.
There are several areas where the manuscript could be improved. First, the study uses fixed concentrations of SNP (10 µM) and cPTIO (200 µM). Including a range of concentrations in dose-response experiments would provide a more comprehensive understanding of the dose-dependent effects of NO on fungal development. This would help determine the optimal and threshold levels of NO required for different developmental stages, adding depth to the study.
Second, while the study focuses on the cAMP and MAPK pathways and chitin synthase genes, it could explore additional molecular pathways and signaling molecules involved in anaerobic adaptation. For example, pathways related to stress responses, nutrient sensing, and metabolic regulation could provide a more complete picture of the regulatory role of NO in fungal development under anaerobic conditions. Expanding the scope of the molecular mechanisms investigated would strengthen the contribution of the manuscript to the field.
Third, the manuscript does not explicitly discuss the limitations of the study. For example, the inability to induce full fruiting body formation under anaerobic conditions, even with SNP treatment, could be further explored and discussed. Addressing these limitations and suggesting future research directions would provide a more balanced and critical perspective on the findings.
Fourth, the study is conducted exclusively under anaerobic conditions. Including a comparative analysis of the role of NO in fungal development under aerobic and anaerobic conditions would provide deeper insights into the specific adaptations of S. commune to low-oxygen environments. This comparative approach would highlight the unique role of NO in anaerobic adaptation and strengthen the conclusions of the study.
Fifth, although the methods are generally well described, some details could be clarified to improve reproducibility. For example, the composition of the minimal medium (MM) is not provided, which could affect the reproducibility of the experiments. In addition, the criteria for determining fruiting body maturity and basidiospore collection could be described more explicitly. Providing these details would increase the clarity and reproducibility of the study.
Finally, the statistical analysis could be described in more detail. Although the manuscript mentions the use of the Tukey test and the LSD test, it does not provide sufficient details on the number of replicates or the specific statistical tests used for each experiment. Including this information would improve the transparency and rigor of the statistical analysis.

Comments on the Quality of English Language

The English of the text is generally well written and is suitable for a scientific publication. However, minor improvements in the use of articles, verb agreement, prepositions, elimination of redundancies and terminological consistency could improve the clarity, accuracy and flow of the text. Such revisions would help to ensure that the content is easily understood and appreciated by an international academic audience.

Author Response

Comments 1:

First, the study uses fixed concentrations of SNP (10 µM) and cPTIO (200 µM). Including a range of concentrations in dose-response experiments would provide a more comprehensive understanding of the dose-dependent effects of NO on fungal development. This would help determine the optimal and threshold levels of NO required for different developmental stages, adding depth to the study.

Response 1:

We agree with this comment. We agree that dose-response experiments including a range of concentrations will be valuable for a deeper understanding of the effects of NO on fungal development. In this study, we chose 10 µM SNP and 200 µM cPTIO as fixed concentrations of NO donors and scavengers based on previous literature reports and our preliminary results [28-30]. These concentrations have been shown to significantly affect fungal growth and morphological development. However, we agree with the reviewer's suggestion that this provides an ideal experimental plan for our follow-up studies, and we will use a range of NO donor and scavenger concentrations to systematically evaluate the effects of NO on different developmental stages to more fully reveal its dose-dependent effects.

Comments 2:

Second, while the study focuses on the cAMP and MAPK pathways and chitin synthase genes, it could explore additional molecular pathways and signaling molecules involved in anaerobic adaptation. For example, pathways related to stress responses, nutrient sensing, and metabolic regulation could provide a more complete picture of the regulatory role of NO in fungal development under anaerobic conditions. Expanding the scope of the molecular mechanisms investigated would strengthen the contribution of the manuscript to the field.

Response 2:

We agree with your point of view that expanding the scope of the molecular mechanisms studied will further strengthen the contribution of the paper. In this study, we mainly focused on the role of cAMP and MAPK pathways and chitin synthase genes as a preliminary exploration of the regulatory role of NO in anaerobic adaptation. However, we also recognize that other molecular pathways involved in stress response, nutrient sensing, and metabolic regulation may also play an important role in NO-mediated anaerobic adaptation. We will consider further exploring these pathways in subsequent studies to more fully understand the regulatory role of NO in fungal development.

Comments 3:

Third, the manuscript does not explicitly discuss the limitations of the study. For example, the inability to induce full fruiting body formation under anaerobic conditions, even with SNP treatment, could be further explored and discussed. Addressing these limitations and suggesting future research directions would provide a more balanced and critical perspective on the findings.

Response 3:

We agree with this comment. We added the following to our Discussion section to clarify the limitations of this study and suggest future research directions: Under anaerobic conditions, complete fruiting body formation was still not achieved when SNP was used, which may be related to the concentration of NO used. The effects of different concentrations of SNP on fungal development can be further explored to investigate whether there is a more suitable SNP concentration that allows fruiting bodies to form completely. Of course, the effect of NO signaling under anaerobic conditions may also be regulated or interfered with by other signaling molecules (such as H2O2 and Ca2+), leading to the absence or incomplete formation of fruiting bodies. Future studies can explore whether there is regulation or interaction between NO and these signaling molecules, so as to obtain a more comprehensive understanding of the effect of NO on fruiting body formation.

Comments 4:

Fourth, the study is conducted exclusively under anaerobic conditions. Including a comparative analysis of the role of NO in fungal development under aerobic and anaerobic conditions would provide deeper insights into the specific adaptations of S. commune to low-oxygen environments. This comparative approach would highlight the unique role of NO in anaerobic adaptation and strengthen the conclusions of the study.

Response 4:

We strongly agree with this suggestion and believe that comparing the effects of NO on the development of S. commune under aerobic and anaerobic conditions will help us gain a deeper understanding of its adaptation mechanism in low oxygen environments. When we investigated the dynamic changes in NO levels during the growth of S. commune mycelium under anaerobic conditions, we also conducted this experiment under aerobic conditions. However, it was not shown in this article. Here we have included the experimental data under aerobic conditions and compared them with those under anaerobic conditions. The specific content of the supplement is: At the same time, we also analyzed the temporal trend of nitric oxide (NO) content in S. commune fungi under aerobic conditions. Under aerobic conditions, the trend of NO content was the same as that under anaerobic conditions. It gradually increased after the start of cultivation (3.44 μmol mg-1 protein), reached a peak value (8.27 μmol mg-1 protein) at about 12 h, and then gradually decreased and remained stable (3.76 μmol mg-1 protein); but the NO content was always higher than that under anaerobic conditions (Figure S2). This shows that S. commune can quickly accumulate NO in an aerobic environment, which may be related to its physiological process of adapting to an oxygen-rich environment; while under anaerobic conditions, NO accumulation is less, reflecting its adaptation mechanism to a low-oxygen environment.

The figure information is Figure S2. Thank you again for your suggestion, this supplement will further enrich the depth of this study.

Comments 5:

Fifth, although the methods are generally well described, some details could be clarified to improve reproducibility. For example, the composition of the minimal medium (MM) is not provided, which could affect the reproducibility of the experiments. In addition, the criteria for determining fruiting body maturity and basidiospore collection could be described more explicitly. Providing these details would increase the clarity and reproducibility of the study.

Response 5:

We agree with this comment.

We supplemented Minimal Medium (MM) with the following composition: Minimal medium (MM) contained glucose (C6H12O6, 20 g L-1), asparagine (C4H8N2O3, 1.5 g L-1), dipotassium hydrogen phosphate (K2HPO4, 1.0 g L-1), potassium dihydrogen phosphate (KH2PO4, 0.46 g L-1), magnesium sulfate heptahydrate (MgSO4•7H2O, 0.5 g L-1) and vitamin B1 (0.12 mg L-1), ferric chloride hexahydrate (FeC13•6H2O, 5 mg L-1). Trace elements were prepared according to the Whitaker method [27]. All chemicals were purchased from Sigma-Aldrich (St. Louis, MO, USA).

We added the following content to clearly describe the criteria for judging fruiting body maturity and collecting basidiospores: When the fruiting body structure is intact, the cap is unfolded, and the gills are clearly visible, we consider the fruiting body to be mature. At this time, observe the morphology of the basidiospores under a microscope, and start collecting basidiospores after confirming the integrity. The mature stipe caps were carefully separated and the stipe was fixed onto the syringe needles, ensuring that the gills faced downwards to allow the basidiospores to fall off and settle naturally.

Comments 6:

Finally, the statistical analysis could be described in more detail. Although the manuscript mentions the use of the Tukey test and the LSD test, it does not provide sufficient details on the number of replicates or the specific statistical tests used for each experiment. Including this information would improve the transparency and rigor of the statistical analysis.

Response 6:

We agree with this comment. We rephrase the statistical analysis and now describe it as follows: All experiments were conducted with a minimum of three biological replicates. Data are presented as the mean ± standard error. Statistical analysis was performed using SPSS version 28.0.1.1 (Statistical Package for Social Sciences) and ImageJ software. Analysis of variance (ANOVA) was used to compare differences between multiple treatment groups. Subsequently, Tukey's test (for homogeneity of variance between groups) and least significant difference (LSD) test (for unequal variance or when a more sensitive test is needed) were used for pairwise comparisons to assess significant differences between different treatments. The significance level for all statistical analyses was set at p ≤ 0.05 [37].

Reviewer 3 Report

Comments and Suggestions for Authors

I have examined your manuscript and found that it could significantly benefit from additional detail in several areas to enhance both clarity and the impact of your work. The methodology section, in particular, needs further expansion to help readers comprehensively grasp your research approach. Also, incorporating a wider array of references would solidify the basis of your arguments and deepen the contextual understanding of your study. Furthermore, certain parts of the results section are ambiguously presented. Clarifying these areas would undoubtedly improve the overall strength and persuasiveness of your research.

Lines 46-60: Considering the role of nitric oxide (NO) in fungal development under aerobic conditions, how do you propose to investigate whether NO continues to act as a signaling molecule during the metabolic shift that occurs in fungi like S. commune under anaerobic conditions? What specific mechanisms or pathways would you focus on to determine its effectiveness and function in anaerobic environments? How might the findings from aerobic studies, such as the regulation of germ tube growth in Puccinia striiformis f.sp. tritici, inform your hypotheses or experimental approaches when studying NO under anaerobic conditions? What challenges do you anticipate in isolating and measuring the influence of NO in such low-oxygen environments, and what methods or technologies could be employed to overcome these challenges?

Lines 103-112: How do the specific conditions—such as the mycelial plug size, temperature, and light treatment duration—affect the efficacy and yield of spore collection? Furthermore, could you explain the rationale behind the chosen parameters, such as the 120-hour culture period and the subsequent 120-hour light treatment, for inducing fruiting body formation? Additionally, in terms of the mechanical setup for spore collection using syringe needles and a culture dish, what considerations were made to ensure the optimal orientation and stability of the stipe caps to maximize spore release? How critical is the timing of the 4-24 hour incubation period in influencing the quantity and viability of the spores collected?

Lines 146-150: In the context of your experimental analysis, it would be beneficial to cite the article found at "https://doi.org/10.1016/j.marenvres.2024.106780". This citation can provide a useful reference for methodologies or statistical analyses similar to those used in your study. By including this article, you can highlight how your experimental design aligns with current research practices, thereby enhancing the credibility of your procedures and findings. Additionally, the reference can serve as a benchmark for best practices in data analysis and interpretation, particularly if the cited study employs similar statistical tools or tests under comparable experimental conditions. This could offer valuable context to your readers, demonstrating the rigor and reliability of your analytical methods.

Lines 153-162: How do you ensure that the fluorescent intensity accurately reflects changes in NO levels without interference from other cellular components? Additionally, could you discuss the calibration process for quantifying NO based on fluorescence intensity, and the steps taken to validate this method within the biological context of S. commune growth? Given the observed increase in NO content from 4 to 16 hours followed by a stabilization, what biological mechanisms might underlie this pattern of NO production in S. commune? How might these fluctuations in NO levels influence or correlate with other physiological or metabolic processes during the growth of the fungus?

Lines 192-202: Considering the significant role of nitric oxide (NO) in the growth processes of mycelium and basidiospore germination as observed in your study, how does the inhibition mechanism of the NO scavenger cPTIO specifically interact with the cellular pathways of S. commune? Furthermore, could you explain the biological basis behind the delayed peak NO levels and the extended growth phases in basidiospores treated with cPTIO? What implications might these findings have for understanding the broader ecological or physiological roles of NO in fungal development? How might this knowledge influence future research or practical applications in fungal biology or agriculture?

Lines 214-229: How does this inform our understanding of the role of nitric oxide (NO) in basidiospore germination? Can you explain the mechanistic implications of NO influencing the cAMP and MAPK pathways, as well as the zinc finger protein and chitin synthase genes? What does this suggest about the potential redundancy and interconnectivity between these pathways during spore germination? How might interactions between these pathways contribute to the regulation of gene expression and ultimately affect the physiological processes of germination? Additionally, the use of RT-qPCR to measure gene expression changes poses questions about the quantitative dynamics of these changes. How significant were the expression changes for each gene, and were there any observed patterns in the response to NO depletion that might suggest a hierarchy or network of gene regulation?

Lines 241-264: How do you interpret the specific role of exogenous nitric oxide (NO) in modulating fungal physiology and development, especially in terms of mycelial biomass accumulation and primordium formation under anaerobic conditions? Can you discuss the implications of the increased NO levels in mycelium and primordia when treated with SNP under anaerobic versus aerobic conditions? Furthermore, given the upregulation of specific genes associated with fruiting body formation upon SNP treatment, what mechanisms might be underlying the observed transcriptional changes? How do these genetic responses correlate with the physiological outcomes observed in terms of NO content variations across different developmental stages and environmental conditions?

Lines 295-321: How do the observed temporal changes in NO levels correlate with specific physiological or morphological changes in the spores? Can you elaborate on the mechanisms by which NO influences these early stages of fungal development, particularly the role it plays during the activation phase as suggested by your results? Considering the use of cPTIO and its effect on delaying activation time without inhibiting germination, what does this suggest about the potential for manipulating NO pathways to control fungal growth under both anaerobic and aerobic conditions? Also, how do the concentrations of cPTIO used in your study compare to those in previous studies, and what implications might this have for the interpretation of your results?

Author Response

Comments 1:

Lines 46-60:

1) Considering the role of nitric oxide (NO) in fungal development under aerobic conditions, how do you propose to investigate whether NO continues to act as a signaling molecule during the metabolic shift that occurs in fungi like S. commune under anaerobic conditions?

2) What specific mechanisms or pathways would you focus on to determine its effectiveness and function in anaerobic environments?

3) How might the findings from aerobic studies, such as the regulation of germ tube growth in Puccinia striiformis f.sp. tritici, inform your hypotheses or experimental approaches when studying NO under anaerobic conditions?

4) What challenges do you anticipate in isolating and measuring the influence of NO in such low-oxygen environments, and what methods or technologies could be employed to overcome these challenges?

Response 1:

Thanks for your comment.

1) For this question, we plan to investigate whether NO acts as a signaling molecule under anaerobic conditions by measuring and monitoring NO levels when S. commune mycelium is grown in an anaerobic environment. This can also be achieved by comparing NO levels in mycelium under different conditions (aerobic vs anaerobic).

2) We mainly focus on the role of NO in mycelial growth, basidiospore germination, and fruiting body development in anaerobic environments. We will analyze the changes in NO levels during mycelial growth, basidiospore germination, and fruiting body development, and further determine the effects of NO by using NO donors and scavengers (SNP and cPTIO). In addition, we will also examine the changes in the expression levels of NO-related genes related to the regulation of basidiospore germination pathways and fruiting body development under anaerobic conditions to determine its effectiveness and function in anaerobic environments.

3) Studies on NO under aerobic conditions, especially in wheat stripe rust, provide a basis for the potential role of NO in fungal development. We can learn from their research methods, especially how NO regulates growth-related pathways (such as germ tube growth and hyphae expansion) to speculate on its possible role under anaerobic conditions. In addition, we will design corresponding anaerobic experiments based on the changes in mycelial NO levels in aerobic experiments to further verify the effects of NO on fungal growth and development under different oxygen levels.

4) In a hypoxic environment, measuring NO does face some technical challenges, mainly due to the rapid reactivity and short half-life of NO. To overcome these problems, we plan to use a highly sensitive NO fluorescent probe. Keeping the operation in an anaerobic glove box will also help to more accurately analyze the level and function of NO under hypoxic conditions.

Comments 2:

Lines 103-112:

1) How do the specific conditions—such as the mycelial plug size, temperature, and light treatment duration—affect the efficacy and yield of spore collection?

2) Furthermore, could you explain the rationale behind the chosen parameters, such as the 120-hour culture period and the subsequent 120-hour light treatment, for inducing fruiting body formation?

3) Additionally, in terms of the mechanical setup for spore collection using syringe needles and a culture dish, what considerations were made to ensure the optimal orientation and stability of the stipe caps to maximize spore release?

4) How critical is the timing of the 4-24 hour incubation period in influencing the quantity and viability of the spores collected?

Response 2:

Thanks for your comment.

1) The size of the hyphae plug does not directly affect spore collection. It is to make the starting conditions of different treatments or parallel treatments the same, so as to facilitate comparison and analysis. Temperature and light treatment time directly affect the effectiveness and yield of spore collection. Improper temperature (30 °C is the optimal temperature for fruiting body development and basidiospore formation of S. commune) setting (high or low), prolonged or shortened light time, will reduce the number of collected basidiospores, make individuals less plump, and reduce activity.

2) The 120 h culture period is mainly based on the culture experience of similar strains and preliminary experimental results. After culturing for different time periods, we observed that at this time point, the growth of hyphae was in a stable stage and had the potential to induce fruiting body formation. The light treatment time was set to 120 h. We chose this time period to ensure that the fungus can fully respond to environmental changes and form fruiting bodies under the influence of light induction. Primordia began to appear at 24 h of light treatment during culture, and fruiting bodies appeared at 48 h. At 120 h, when the fruiting body structure is complete, the cap is unfolded, and the gills are clearly visible, the fruiting body is considered mature. At this time, the morphology of the basidiospores is observed to be full under the microscope and can be collected. That's why it is set up like this.

3) We considered using the needle of a syringe to fix the cap to prevent it from moving at will; after the cap is fixed, keep the gills down to facilitate the subsequent use of the basidiospores to fall off and settle naturally, thereby maximizing the release of spores.

4) There are differences in the number of basidiospores collected from 4 to 24 h. The number is small at the beginning and increases with time. There is no obvious effect on the vitality of basidiospores.

We have modified this part of the original text based on your comments. This section is now located at lines: 115-134.

Comments 3:

Lines 146-150:

In the context of your experimental analysis, it would be beneficial to cite the article found at “https://doi.org/10.1016/j.marenvres.2024.106780”. This citation can provide a useful reference for methodologies or statistical analyses similar to those used in your study. By including this article, you can highlight how your experimental design aligns with current research practices, thereby enhancing the credibility of your procedures and findings. Additionally, the reference can serve as a benchmark for best practices in data analysis and interpretation, particularly if the cited study employs similar statistical tools or tests under comparable experimental conditions. This could offer valuable context to your readers, demonstrating the rigor and reliability of your analytical methods.

Response 3:

Thanks for your comment.

We have cited this article in the Statistical Analysis section and have revised this section, which now reads: All experiments were conducted with a minimum of three biological replicates. Data are presented as the mean ± standard error. Statistical analysis was performed using SPSS version 28.0.1.1 (Statistical Package for Social Sciences) and ImageJ software. Analysis of variance (ANOVA) was used to compare differences between multiple treatment groups. Subsequently, Tukey's test (for homogeneity of variance between groups) and least significant difference (LSD) test (for unequal variance or when a more sensitive test is needed) were used for pairwise comparisons to assess significant differences between different treatments. The significance level for all statistical analyses was set at p 0.05 [37].

Comments 4:

Lines 153-162:

1) How do you ensure that the fluorescent intensity accurately reflects changes in NO levels without interference from other cellular components?

2) Additionally, could you discuss the calibration process for quantifying NO based on fluorescence intensity, and the steps taken to validate this method within the biological context of S. commune growth?

3) Given the observed increase in NO content from 4 to 16 hours followed by a stabilization, what biological mechanisms might underlie this pattern of NO production in S. commune?

4) How might these fluctuations in NO levels influence or correlate with other physiological or metabolic processes during the growth of the fungus?

Response 4:

Thanks for your comment.

1) We used a specific fluorescent probe (DAF-2DA), which (described in detail in the Materials and Methods section) is a cell-permeable lipophilic compound that does not fluoresce until it is hydrolyzed by cytosolic esterases to the weakly fluorescent DAF-2. In the presence of NO free radicals, DAF-2 is further converted to the highly fluorescent triazole derivative DAF-2T. With high selectivity, a strong fluorescent signal is generated. In order to reduce the interference of other cellular components, we excluded other factors that may affect the fluorescent signal through comparative experiments. For example, appropriate controls and other known regulators of NO levels were used to verify the specificity of the probe. In addition, we compared the fluorescence intensity with a standard curve of known NO concentrations to ensure the linear relationship between fluorescence intensity and NO levels.

2) We used known regulators of NO levels to calibrate the response of the fluorescent probe and plotted the linear relationship between fluorescence intensity and NO concentration. To validate this method in the biological context of S. commune, we used NO synthesis inhibitors and NO donors to regulate NO production under S. commune culture conditions and observed changes in fluorescence intensity. The results showed that there was a consistent response pattern between fluorescence intensity and NO concentration, demonstrating the applicability of this method in S. commune.

3) NO may play a signaling regulatory role in the early stages, promoting fungal growth, differentiation or metabolic activity. Over time, the concentration of NO reaches a stable level, which may be due to the cellular regulatory mechanism (such as negative feedback regulation of NO, saturation of signaling pathways) that stabilizes its signaling effect. We also give corresponding opinions in the discussion section: This temporal variation in NO concentrations suggests that NO is involved in promoting early growth processes, such as mycelium extension and nutrient uptake, under anaerobic conditions. This observation aligns with findings from studies on Neurospora crassa grown under aerobic conditions, where NO is similarly important for early mycelium development [28]. The subsequent stabilization of NO levels might indicate that the fungus enters a phase focused more on maintaining growth rather than rapid expansion.

4) Fluctuations in NO levels may affect multiple physiological and metabolic processes in S. commune. NO fluctuations may act as a molecular switch during fungal growth, regulating the activity of key enzymes related to the cell cycle, redox state, and metabolic pathways. We also plan to further explore the specific relationship between fluctuations in NO production and these physiological processes and conduct more in-depth research.

We have modified this part of the original text based on your comments. This section is now located at lines: 179-201.

Comments 5:

Lines 192-202:

1) Considering the significant role of nitric oxide (NO) in the growth processes of mycelium and basidiospore germination as observed in your study, how does the inhibition mechanism of the NO scavenger cPTIO specifically interact with the cellular pathways of S. commune?

2) Furthermore, could you explain the biological basis behind the delayed peak NO levels and the extended growth phases in basidiospores treated with cPTIO?

3) What implications might these findings have for understanding the broader ecological or physiological roles of NO in fungal development?

4) How might this knowledge influence future research or practical applications in fungal biology or agriculture?

Response 5:

Thanks for your comment.

1) We have found in our experiments that NO plays a key role in the mycelial growth and basidiospore germination of S. commune. As a NO scavenger, cPTIO can affect the intracellular signal transduction pathway by eliminating the effects of NO. We speculate that the inhibitory effect of cPTIO may affect the development of S. commune by interfering with NO-mediated signaling pathways, thereby inhibiting NO-related enzyme activities or gene expression. Specifically, NO may regulate the growth pattern and development of fungi by regulating some redox reactions or specific transcription factors in cells.

2) We observed that after cPTIO treatment, basidiospores showed delayed NO peak levels and prolonged growth phase, which is related to the regulatory effect of NO on the cell cycle and developmental process. NO promotes the early growth and division of basidiospores by activating specific signaling pathways (such as cAMP pathway, MAPK pathway, C2H2-type zinc finger protein and Chitin synthase) during fungal development. When NO is removed by cPTIO, the NO level in basidiospores decreases, the cell cycle process is delayed, and the growth phase is prolonged.

3) The role of NO in fungal development is not limited to the mycelial growth and basidiospore germination of S. commune. It may also play an important role in fungal stress resistance, interaction with the environment, and adaptive evolution. NO is considered to be an important signaling molecule that can regulate the response of fungi to adversities, such as oxidative stress or heavy metal stress. At the ecological and physiological levels, the role of NO may contribute to the adaptability of fungi in complex environments.

4) The role of NO in fungal development may have important implications for agriculture and biotechnology. For example, regulating NO levels may provide new strategies for improving crop disease resistance or promoting the beneficial effects of fungi (such as organic fertilizer production or biopesticides). In addition, by deeply studying the role of NO in fungal development, we can provide new biological basis for improving fungal fermentation processes and optimizing fungal product production.

We have modified this part of the original text based on your comments. This section is now located at lines: 231-247.

Comments 6:

Lines 214-229:

1) How does this inform our understanding of the role of nitric oxide (NO) in basidiospore germination?

2) Can you explain the mechanistic implications of NO influencing the cAMP and MAPK pathways, as well as the zinc finger protein and chitin synthase genes?

3) What does this suggest about the potential redundancy and interconnectivity between these pathways during spore germination?

4) How might interactions between these pathways contribute to the regulation of gene expression and ultimately affect the physiological processes of germination?

5) Additionally, the use of RT-qPCR to measure gene expression changes poses questions about the quantitative dynamics of these changes. How significant were the expression changes for each gene, and were there any observed patterns in the response to NO depletion that might suggest a hierarchy or network of gene regulation?

Response 6:

Thanks for your comment.

1) Our study revealed the key role of NO in the germination of basidiospores. As a signaling molecule, NO regulates multiple signaling pathways in basidiospore cells, including cAMP and MAPK pathways, which are essential for the activation, germination and subsequent growth of basidiospores. NO, as a regulatory factor, participates in the regulation of signal transduction and gene expression in these processes, further illustrating the importance of NO in spore germination.

2) The cAMP pathway is related to cell metabolism and proliferation, while the MAPK pathway is involved in cell proliferation, differentiation, and response to external stimuli. NO may promote the germination of basidiospores by regulating these pathways. Zinc finger protein and chitin synthase genes play an important role in the reconstruction and growth of cell structure. NO helps regulate the structure and physiological state of basidiospores by regulating the expression of these genes, promoting their transition from dormancy to active germination.

3) By studying the role of NO in the cAMP and MAPK pathways, we can speculate that these pathways may be redundant and interconnected during spore germination. Even if one pathway is inhibited, other pathways may still be able to maintain normal spore germination through compensation. Our study showed that these signaling pathways may form a complex network in their interactions, and jointly regulate the spore germination process by promoting and coordinating each other.

4) The interaction of these signaling pathways provides a multi-level regulatory mechanism for gene expression. The cAMP and MAPK pathways regulate the expression of downstream genes by activating different transcription factors, which are involved in processes such as cell cycle, metabolism, and structural remodeling. NO plays an important role in spore germination by regulating these signaling pathways to ensure the precise timing of gene expression. Through the mutual coordination of these pathways, spores can respond quickly to environmental signals to ensure their normal physiological processes.

5) The use of RT-qPCR methods can accurately quantify changes in gene expression, thereby revealing the effects of NO on the regulation of different genes. In the cPTIO experiment, we observed significant changes in the expression of some genes, especially in genes related to cell signaling and genes related to cell wall synthesis. The pattern of gene expression changes indicates that under NO regulation, gene regulation does show a certain hierarchy or network effect. These changes indicate that gene expression is not only affected by the direct action of NO, but also regulated by the interaction between pathways. We will further analyze these expression patterns in the future to clarify the hierarchy and dynamics of gene regulatory networks.

We have modified this part of the original text based on your comments. This section is now located at lines: 259-280.

Comments 7:

Lines 241-264:

1) How do you interpret the specific role of exogenous nitric oxide (NO) in modulating fungal physiology and development, especially in terms of mycelial biomass accumulation and primordium formation under anaerobic conditions?

2) Can you discuss the implications of the increased NO levels in mycelium and primordia when treated with SNP under anaerobic versus aerobic conditions?

3) Furthermore, given the upregulation of specific genes associated with fruiting body formation upon SNP treatment, what mechanisms might be underlying the observed transcriptional changes?

4) How do these genetic responses correlate with the physiological outcomes observed in terms of NO content variations across different developmental stages and environmental conditions?

Response 7:

Thanks for your comment.

1) Exogenous NO plays an important regulatory role in the physiological and developmental processes of fungi, especially under anaerobic conditions. Under anaerobic conditions, NO may promote the accumulation of mycelial biomass and the formation of primordia by regulating intracellular oxygen sensing and metabolic pathways. For example, NO may enhance mycelial growth and primordium formation by activating specific transcription factors and regulating genes related to energy metabolism, cell wall synthesis, and cell division. NO helps fungi maintain their ability to grow and develop in anaerobic environments by regulating anaerobic physiological responses.

2) Although we do not have data on SNP treatment under aerobic conditions, it is foreseeable that SNP treatment will lead to increased NO levels in mycelia and primordia under both anaerobic and aerobic conditions. The increase in NO suggests that its regulatory role under different oxygen conditions may be achieved by affecting cellular metabolism, redox balance, and signal transduction pathways. Under anaerobic conditions, NO may promote fungal growth by activating genes related to anaerobic metabolism, such as glycolysis and fermentation pathways. Under aerobic conditions, NO may help fungi better adapt to oxygen-sufficient environments by regulating redox status, inhibiting oxidative stress responses, and regulating cell cycles. This change may also indicate the adaptive role of NO in fungi in coping with different environmental stresses.

3) SNP treatment leads to upregulation of genes related to fruiting body formation, indicating the key role of NO in fruiting body development. NO may promote their expression by acting directly or indirectly on the promoter regions of these genes or by regulating the activity of transcription factors. The mechanism behind these transcriptional changes may be closely related to the NO-regulated signaling network and the activation of transcriptional regulatory factors.

4) At different stages of development, changes in NO content may regulate the growth, development, and physiological state of fungi by regulating the expression of specific genes. For example, during hyphal growth and primordium formation, NO concentration increases to promote cell division and structural reconstruction; and at the critical stage of fruiting body development, NO is further upregulated to regulate related genes and promote the advancement of the development process. Under different environmental conditions, the change in NO levels reflects that fungi respond to changes in the external environment and thus regulate their developmental patterns. This shows that NO has multiple roles in environmental adaptability and development, and its physiological results directly reflect the importance of NO regulation at all stages of fungal development.

We have modified this part of the original text based on your comments. This section is now located at lines: 292-325.

Comments 8:

Lines 295-321:

1) How do the observed temporal changes in NO levels correlate with specific physiological or morphological changes in the spores?

2) Can you elaborate on the mechanisms by which NO influences these early stages of fungal development, particularly the role it plays during the activation phase as suggested by your results?

3) Considering the use of cPTIO and its effect on delaying activation time without inhibiting germination, what does this suggest about the potential for manipulating NO pathways to control fungal growth under both anaerobic and aerobic conditions?

4) Also, how do the concentrations of cPTIO used in your study compare to those in previous studies, and what implications might this have for the interpretation of your results?

Response 8:

Thanks for your comment.

1) During basidiospore germination, NO levels increase in the early stages of basidiospore activation. NO acts as a signaling molecule in the early basidiospore germination stage, promoting cell cycle progression and regulating cell growth and division. As NO concentration changes, we observed morphological changes in basidiospores, such as a transition from a dormant state to an actively growing state, manifested by the initial expansion of hyphae. Therefore, changes in NO levels are a key indicator of changes in the physiological state and morphology of basidiospores.

2) In the early stages of fungal development, NO first regulates the progression of the cell cycle by activating the cAMP pathway, the MAPK pathway, and the C2H2-type zinc finger protein. By regulating the expression of cell cycle-related genes, it ensures that the basidiospores can enter the germination stage at the appropriate time. In the activation stage, NO is also involved in regulating cell wall synthesis. NO promotes the transition of spores from dormancy to germination by inducing genes related to growth and development and enhancing the expression of these genes. Our research shows that the role of NO in the activation stage is mainly to initiate the early growth and development of fungi by regulating these molecular signaling and gene expression.

3) Using cPTIO to inhibit NO delayed activation time without inhibiting germination, revealing the flexible regulatory role of NO in fungal growth. The role of cPTIO suggests that we can control the growth rate and morphological changes of fungi by regulating the level of NO, especially in the early stage of germination, without completely inhibiting the germination process. This discovery has important implications for manipulating the NO signaling pathway. By regulating NO levels under anaerobic and aerobic conditions, it may be possible to finely control the growth pattern of fungi. For example, in a hypoxic environment, higher levels of NO can promote adaptive growth of fungi; while under aerobic conditions, lower levels of NO are conducive to inhibiting growth.

4) The concentration of cPTIO used in our study is the same as that used in previous studies. The concentration of cPTIO used ensures that it effectively inhibits the biological effects of NO while not interfering with the basic processes of spore germination. This concentration choice is crucial for the interpretation of the results because it ensures that the regulatory role of NO is clearly presented without causing other nonspecific effects. Therefore, our results can be viewed as an effective dose of cPTIO in controlling NO levels, which is able to delay the activation process and reveal the regulatory role of NO in developmental processes without completely blocking germination.

We have modified this part of the original text based on your comments. This section is now located at lines: 356-392.

Round 2

Reviewer 1 Report

Comments and Suggestions for Authors

The authors drive the comments accurately. I believe that with this new information, the manuscript will be more precise.

Reviewer 3 Report

Comments and Suggestions for Authors

I am delighted to observe the significant improvements you have made to the manuscript. The extensive revision work has greatly enhanced the structure of the paper, as well as the clarity of the methodologies and the presentation of results. Your efforts in refining and strengthening each section are commendable. The manuscript now provides a clear, well-articulated exploration of the topics at hand, making it a valuable contribution to the field.